# The Gut–Vascular Barrier as a New Protagonist in Intestinal and Extraintestinal Diseases

**DOI:** 10.3390/ijms24021470

**Published:** 2023-01-12

**Authors:** Natalia Di Tommaso, Francesco Santopaolo, Antonio Gasbarrini, Francesca Romana Ponziani

**Affiliations:** 1Internal Medicine and Gastroenterology, Fondazione Policlinico Universitario Agostino Gemelli IRCCS, 00168 Rome, Italy; 2Translational Medicine and Surgery Department, Università Cattolica del Sacro Cuore, 00168 Rome, Italy

**Keywords:** GVB: gut–vascular barrier, PVB: choroid plexus vascular barrier, neuroinflammation, endothelium, PV-1: plasmalemma vesicle-associated protein-1

## Abstract

The intestinal barrier, with its multiple layers, is the first line of defense between the outside world and the intestine. Its disruption, resulting in increased intestinal permeability, is a recognized pathogenic factor of intestinal and extra-intestinal diseases. The identification of a gut–vascular barrier (GVB), consisting of a structured endothelium below the epithelial layer, has led to new evidence on the etiology and management of diseases of the gut–liver axis and the gut–brain axis, with recent implications in oncology as well. The gut–brain axis is involved in several neuroinflammatory processes. In particular, the recent description of a choroid plexus vascular barrier regulating brain permeability under conditions of gut inflammation identifies the endothelium as a key regulator in maintaining tissue homeostasis and health.

## 1. Introduction

The intestinal barrier is a structural, functional and immunological defense against external factors. It is a multilayered structure, consisting of the mucus, epithelium, and lamina propria [1], and above them lies another key component, the gut microbiota [2,3].

The mucus layer is made of water and mucins secreted by goblet cells, exerts antimicrobial properties, and keeps bacteria distant from the mucosa [4,5,6].

Goblet cells, enterocytes, enteroendocrine cells, Paneth cells and microfold cells are part of the epithelial layer [7]. Tight junctions (TJs), adherens junctions (AJs), and desmosomes located at intercellular contact points of intestinal epithelial cells (IECs) regulate the selective passage of molecules from the intestinal lumen [8]. They are dynamic specialized structures made of transmembrane proteins, such as claudins, occludin, junctional adhesion molecules (JAM), tricellulin, angulins, and of intracellular proteins anchored to the actin cytoskeleton, such as zonula occludens (ZO); their interaction finely tunes the intestinal epithelial barrier (IEB) function [9,10].

Intestinal immune homeostasis is maintained by the interaction between IECs and the gut microbiota through pattern-recognition receptors (PRRs), such as toll-like receptors (TLRs) and NOD-like receptors (NLRs), which activate the innate immune response [7].

In healthy conditions, a small amount of bacteria crosses the intestinal lumen [11]; IECs are able to discriminate commensals from pathogens, regulating PRR expression to prevent excessive inflammatory response [12]. Microbiota in turn stimulates the immune system to recognize pathogens, and at the same time limits mucosal damage during inflammation, playing an immunoregulatory role [13,14].

However, in the case of immune system dysfunction, intestinal barrier damage and/or dysbiosis, there is a break in this homeostatic balance, and an increased amount of bacteria and their products translocate through the IEB, reaching mesenteric lymph nodes and systemic circulation [11,15,16]. Bacterial translocation (BT) boosts mucosal and systemic inflammation, further increasing intestinal permeability in a vicious circle [17]. This mechanistic model of gut-driven inflammation is a well-recognized contributor to the onset and progression of metabolic, inflammatory and liver diseases, as well as cancer [18,19,20,21].

The fine-tuning of IEB permeability is critical to prevent external stressors from reaching the lamina propria, which hosts immune cells and is rich in blood vessels, therefore being the optimal environment for the activation of a systemic inflammatory response [22,23].

Recent studies revealed the existence of an additional layer in the intestinal barrier, the “gut–vascular unit”, which is critical in maintaining its homeostasis. The objective of this review is to analyze its functioning and the evidence on its dysregulation in gastrointestinal and systemic diseases.

## 2. The Gut–Vascular Barrier

Great emphasis has recently been placed on gut endothelial cells (ECs) as an additional element in regulating the intestinal barrier [24]. ECs, like enterocytes, are connected by AJs, TJs, catenin and cadherin proteins, and play an important role in regulating vessels’ permeability [25,26].

Intestinal ECs are fenestrated, which means they host small pores delimited by a fe-nestral diaphragm [27,28,29]. Fenestral diaphragm formation requires the presence of an endothelial membrane glycoprotein, called the plasmalemma vesicle-associated protein-1 (PV-1), encoded by the PLVAP gene [28]. PV-1 is an essential regulator of endothelial homeostasis and permeability [29]; as a confirmation of this important role, PLVAP gene mutations are associated with a severe protein-losing enteropathy in vivo [30,31].

Additionally, in the condition of increased endothelial leakage and vascular damage, PV-1 upregulation is detected at immunohistochemistry analysis [32].

ECs also participate in mucosal immunology and express TLRs [33], adhesion molecules such as E-selectin, vascular cell adhesion molecule 1 (VCAM-1), and intercellular adhesion molecule 1 (ICAM-1), especially in the case of gut inflammation [34,35]. Furthermore, ECs form together with pericytes and enteric glial cells with an additional layer underneath the intestinal epithelium named the gut–vascular barrier (GVB) [36,37].

The GVB has several similarities with the blood–brain barrier (BBB), including the incremental expression of PV-1 during damages and the regulation by the Wnt/β-catenin signaling pathway [32,38,39,40].

The earliest data uncovering the pathophysiology of the GVB were published by Spadoni et al. [36]; the study demonstrated that mice orally infected by *Salmonella typhimurium* developed a systemic disease secondary to GVB disruption, as documented by the incremental expression of PV-1, similarly to conditions of altered BBB integrity [32,38]. While in normal conditions, GVB permeability was restricted to 4 kDa molecules, during *Salmonella* infection, the leakage of 70 kDa fluorescein isothiocyanate (FITC)–dextran was observed, confirming an increased vascular leakiness. GVB impairment was associated with derangement of the Wnt/β-catenin signaling, since mutant mice expressing a β-catenin resistant to degradation did not develop bacterial dissemination. Finally, the authors observed a similar mechanism of increased PV-1 expression in the human gut of celiac individuals who presented hypertransaminasemia despite adherence to a gluten-free diet, suggesting that a damaged GVB may be the cause of liver injury in that setting [36]. Further investigations showed that *Salmonella* infection was not associated with inflammatory gene activation, but rather with increased expression of epithelial/endothelial to mesenchymal transition genes, and genes involved in angiogenesis and the bile acids metabolism [37]. During the endothelial to mesenchymal transition (EndMT), transforming growth factor β (TGF-β) and pro-inflammatory cytokines induce ECs to lose their endothelial markers and acquire mesenchymal cells’ features, resulting in endothelial dysfunction [41,42]. EndMT is described in several models of tissue fibrosis, cardiovascular diseases, and cancer [43,44], and is also promoted by bacteria [45,46]. Thus, in their subsequent analysis [37], Spadoni et al. proposed a bacteria-driven dysregulation of the GVB based on EC reprogramming, showing that *Salmonella* infection could enhance the expression of mesenchymal genes and EndMT markers in ECs, resulting in vascular remodeling.

Following these preliminary findings, the GVB is now considered an integral component of the intestinal barrier, and its dysfunction has emerged as a concurrent cause of intestinal and extraintestinal diseases [24].

Figure 1 summarizes the main features of a healthy or impaired GVB.

## 3. Gut–Vascular Barrier in Liver Disease

The integrity of the intestinal epithelium is a basic requirement for homeostatic balance between the gut and the liver, which are interconnected anatomically by vascular and biliary structures that realize the so-called gut–liver axis [47,48].

Changes in gut microbiota composition, high-fat diet (HFD), genetic predisposition, drugs and external toxins can compromise IEB integrity, leading to TJ post-transcriptional alterations and favoring the translocation of pathogens and toxins into the portal system, mesenteric lymph nodes and systemic circulation, triggering an inflammatory response mediated by increased lipopolysaccharide (LPS) serum levels and consequent endotoxemia [49,50].

After the acknowledgment of the key role of the GVB in intestinal physiology, there has been growing evidence on its involvement in liver disease, mainly nonalcoholic fatty liver disease (NAFLD) and alcoholic liver disease [48].

### 3.1. Nonalcoholic Fatty Liver Disease and Nonalcoholic Steatohepatitis

NAFLD includes a spectrum of alterations that range from simple fat accumulation (nonalcoholic fatty liver, NAFL) to inflammatory damage (nonalcoholic steatohepatitis, NASH) and the development of cirrhosis with its complications [51].

NAFLD always coexists with metabolic diseases, especially obesity and type 2 diabetes mellitus (T2DM), as insulin-resistance is a main driver of fatty acids accumulation in the liver and their oxidation, causing liver inflammation, Kupffer and stellate cells activation, and deposition of fibrotic tissue until the development of cirrhosis [52]. NAFLD development is strongly associated with obesity and a high-sugar and -fat diet, that contribute to liver and adipose tissue lipotoxicity [53,54]. Fructose-rich diets have been associated with gut–liver axis impairment, through the downregulation of TJs and changes in the gut microbiota composition, leading to endotoxemia and liver inflammation, with the activation of Kupffer cells and hepatic stellate cells [55,56,57]. It has been recently demonstrated that fructose effects on TJs and liver function could be explained by the induction of cytochrome P450-2E1 (CYP2E1), resulting in an increased expression of reactive oxygen species [58].

Additionally, the Western diet is associated with systemic endotoxemia in both human and animal studies [59,60,61,62]; a high-fat diet (HFD) directly affects the intestinal barrier in mice models, reducing zonula occludens-1 (ZO-1) expression, whereas the administration of antibiotics reduces systemic inflammation and LPS levels [63]. A study also demonstrated two different patterns of metabolic alterations in mice fed with HFD: the responder group had a significant increase in fasting glycemia, insulinemia and insulin resistance, while this was not observed in non-responders. Fecal microbiota transplantation (FMT) from responders into germ-free (GF) recipient mice reproduced metabolic alterations, leading to insulin resistance and liver steatosis [64], confirming previous evidence that GF mice are resistant to the metabolic consequences of HFD [65].

These studies suggest that dietary modulation of the gut microbiome may affect the intestinal barrier, and dysbiosis has been demonstrated in several animal and human studies to be associated with NAFLD and NASH [66,67,68,69].

Additionally, probiotics administration could help reduce liver inflammation and endotoxemia through upregulation of TJs [70].

Diet and dysbiosis can also be implicated in liver damage through the alteration of the GVB. As previously documented in a mouse model of NAFLD, HFD plus the administration of dextran sodium sulfate (DSS) to induce colitis was able to exacerbate liver inflammation and fibrosis compared to DSS alone [71]. DSS damage on intestinal epithelium led to the downregulation of ZO-1 and claudin-1 expression, which was not observed in HFD-fed-only mice, confirming that intestinal barrier impairment is a mandatory step for the development of liver damage [71]. Moreover, in DSS plus HFD mice, endothelial permeability was higher than in DSS mice fed a normal diet, as demonstrated by the detection of fluorescein isothiocyanate (FITC)-dextran 70 KDa in serum, liver and spleen 1 h after intestinal injection; colonic microvascular expression of PV-1 was also increased [71]. Thus, these preliminary data demonstrate that the disruption of the IEB and GVB is necessary to induce HFD-related liver injury in NAFLD animal models. However, HFD itself could be responsible for intestinal barrier damage. Mouries et al. showed that mice fed with HFD presented sequential damage in the IEB and GVB, as documented by the early loss (within 48 h) of epithelial ZO-1, and the subsequent increase (after 1 week) of PV-1 expression by ECs [72]. The initial disruption of the IEB allowed bacterial translocation in the lamina propria, which further impaired the GVB leading to bacterial translocation in the liver parenchyma. All these alterations were demonstrated to precede liver injury and the development of insulin resistance. The observation was elegantly confirmed by the increase in PV-1 expression after FMT from HFD-fed mice into recipient mice. Finally, the constitutive activation of β-catenin signaling or the upregulation of β-catenin target genes by obeticholic acid (OCA), a strong agonist of the farnesoid X receptor (FXR), could prevent GVB disruption and bacterial translocation to the liver [72]. This pivotal study highlighted a central role of diet-induced dysbiosis in NASH pathogenesis; even if the authors demonstrated that PV-1 expression was enhanced in colonic tissue specimens obtained from nine patients with NASH, these data deserve to be further addressed in larger populations of patients [72].

As described in the aforementioned studies, dysbiosis and unhealthy diets heavily influence the intestinal barrier function, acting in a sequential manner on the epithelial layer and the endothelial layer, as a sort of “first” and “second” hit. Dietary interventions, and even probiotics administration, which could further help to achieve a beneficial bile acids composition, have been extensively analyzed in NAFLD treatment [73,74]. A double therapeutic strategy based on gut microbiota modulation through diet and FXR agonists, able to reinforce both epithelial and endothelial layer, is appealing. However, trials investigating the use of FXR agonists in patients with NAFLD did not show a significant attenuation of inflammatory liver injury and fibrosis; furthermore, these drugs can exert unfavorable effects on lipid metabolism. Thus, preliminary evidence from mice models needs to be further addressed by human studies [75].

### 3.2. Alcoholic Liver Disease

In alcohol-associated liver disease (ALD), liver toxicity is mediated by direct effects of ethanol and its metabolite acetaldehyde, but also by their indirect effect on the gut microbiome and intestinal barrier function [76,77,78,79].

In fact, alcohol is associated with modification in gut microbiota composition, which mainly consists in the decrease in *Ruminococcaceae* [80], the increase in *Bacteroides*, and the reduction in *Akkermansia*, featuring a dysbiotic and pro-inflammatory environment [81,82]. Furthermore, decrease in gut mycobiome diversity, with the overgrowth of *Candida* spp. producing the endotoxin candidalysin, favors liver damage and is associated with the severity of liver impairment and a worse patients’ outcome [83,84,85].

Alcohol is predominantly metabolized by the liver into acetaldehyde and then acetate, and acetate levels increase in the blood and gut after alcohol consumption [86,87]. As recently demonstrated, the effects of alcohol on the composition of the gut microbiota could not be related to direct microbial metabolism of ethanol, but, rather, could be an indirect consequence of bacterial adaptation to increased levels of acetate, which can be a source of energy for bacteria [88]. This study points out new considerations about alcohol-induced dysbiosis, that could strongly influence future studies.

Ethanol also disassembles TJ’s structure, triggering bacterial translocation and endotoxemia [89,90]. One of the mechanisms behind this effect involves the immune system, through the downregulation of two antimicrobial proteins, the regenerating islet-derived 3 beta (Reg3b) and gamma (Reg3g) lectins [91,92]. This is a downstream result of the reduced conversion of tryptophan in indole metabolites by the gut microbiota, including the indole-3-acetic acid (IAA), which is a ligand of the aryl-hydrocarbon receptor (AHR). In gut immune cells, a defective AHR pathway impairs the production of interleukin 22 (IL22), which is pivotal for the maintenance of gut homeostasis through Reg3g upregulation [93,94].

Recently, an increased expression of PV-1 has been reported in a population of patients presenting alcohol use disorder (AUD), addressing vascular leakiness as a concurrent mechanism of bacterial translocation in ALD [95]. *Akkermansia* administration was able to reduce by 47% PV-1 ileal expression in mice on the ethanol-containing Lieber–DeCarli diet, reinforcing the GVB, although this result was not statistically significant [96]. The administration of the same ethanol diet in β-catenin gain-of-function mice resulted in liver injury as in control mice, suggesting the existence of other alcohol-related mechanisms involved in GVB disruption [96]. Thus, understanding the mechanisms behind alcohol selection of a peculiar microbiota composition could help counteract alcohol-induced dysbiosis and reduce liver damage.

### 3.3. Cirrhosis

Cirrhosis is the paradigm of intestinal barrier impairment. Portal hypertension, changes in the gut microbiota composition, gastrointestinal dysfunction and immune system defects contribute altogether to mucosal damage and systemic inflammation [97,98,99,100,101]. The translocation of a huge amount of bacteria and their products to the liver and systemic circulation leads to a pro-inflammatory response, resulting in hepatic fibrogenesis consequent to hepatic stellate cells and Kupffer cells activation [102].

In a murine model of cirrhosis, GVB impairment was demonstrated to be independent of the gut microbiota composition, as it was observed in both germ-free (GF) animals and control mice. Differently from mice with pre-hepatic portal hypertension (PPVL), those subjected to bile duct ligation or treated with CCl4- to induce cirrhosis showed a reduced number of goblet cells and mucus thickness in the small intestine, with concomitant bacterial overgrowth in the inner mucus layer. Cirrhotic mice showed translocation of GFP-*E.coli* and interepithelial leakage of FITC-dextran 70 kDa, along with the downregulation of the TJ proteins ZO-1, occludin and claudin, which was not observed in control or PPVL mice. This was paralleled by GVB disruption, as documented by increased expression of PV-1 and extravasation of large molecules such as FITC-dextran 70 kDa and 150 kDa in the lamina propria early after intravenous injection. Oral administration of OCA or fexaramine (Fex), two FXR agonists, ameliorated GFP-*E.coli* translocation in cirrhotic mice; although both of them increased the expression of ileal TJ proteins, only OCA presented systemic absorption and could reinforce the GVB [103]. A profound alteration in a bile acids pool characterizes liver cirrhosis [104,105]; therefore, the benefit of FXR agonists in ameliorating epithelial or GVB integrity may be correlated with the restoration of impaired bile acids signaling.

Intriguingly, Takeda G-protein-coupled receptor 5 (TGR5) agonists or the FXR agonist PX20606 have been demonstrated to reinforce the endothelial barrier, relax liver sinusoids and reduce pre-hepatic and intrahepatic portal hypertension in animal models [106,107], but further studies are needed to validate this evidence in human subjects.

## 4. Gut–Vascular Barrier in Colorectal Cancer Progression

The disruption of the GVB may also be implicated in the development of liver metastases from colorectal cancer (CRC). Tumor-associated microbiota could drive the formation of a pre-metastatic niche (PMN) via GVB impairment [108]. PMN is a biochemical, anatomical and immunological environment driven by the primary tumor that helps seeding of metastatic cells at distant sites. It is made of cytokines, growth factors and immune cells, realizing a favorable milieu for metastatic cells [109,110,111]. Endothelial derangement is critical for the PMN formation [112], and the role of the gut microbiota in this process has been well-documented. In fact, HFD-induced dysbiosis favors PMN development in the lung through the activation of nuclear factor-κB (NF-κB) signaling and the release of cytokines such as tumor necrosis factor-α (TNF-α) and C-C chemokine ligand 2 (CCL2) by M1-macrophages, which are able to promote tumor progression [113,114,115]. In accordance with previous studies [116,117], counteracting dysbiosis and M1-macrophage signaling by the administration of glycyrrhizic acid prevented PMN formation [113].

In a recent study, Bertocchi et al. [108] conducted a retrospective analysis comparing colic specimens from 179 patients with resected CRC and 10 healthy individuals, showing a tendency to develop metachronous metastases in patients with higher expression of PV-1 in CD31 + ECs. PV-1 expression was also associated with a lower rate of progression-free survival, and was recognized as an independent prognostic factor of cancer recurrence. A higher PV-1 expression in patients with metastatic CRC correlated with an increased bacterial colonization of liver metastatic lesions. The authors also demonstrated in a mouse model of CRC that reducing liver bacteria colonization by the administration of antibiotics was paralleled by the decrease in pro-inflammatory cytokines’ expression and immune cells’ recruitment, supporting the hypothesis that the gut microbiota participates in liver PMN formation. A spatial connection between the microbial communities of CRC and the liver was also confirmed, with a gradient of *E. coli* representation. In particular, the *E. coli* C17 strain isolated from CRC lesions was able to trigger GVB damage and bacterial translocation to the liver, driving PMN onset; this was correlated with the virulence factors Virf1 and 2 that are involved in the formation of the type III secretion system (TTSS) machinery, which has been already demonstrated to be linked to GVB impairment during *Salmonella* infection [36]. Interestingly, *E. coli* C17 was also documented in CRC tissue and liver metastases of CRC patients with higher expression of PV-1 [108]. Figure 2A shows the main mechanisms involved in CRC progression in the presence of altered GVB.

These data highlight the implications of GVB in the mechanisms of tumor progression, and that tumor-associated microbiota is pivotal in this process; the results are even more interesting given that administration of antibiotics before the onset of metastases in mice was able to reduce the expression of PV-1 in the primary tumor as well as the number of liver lesions [108], opening the field also to interventional studies.

It was also shown that the exposure of colonic short-chain fatty acids (SCFAs) to *E. coli* strains isolated from CRC patients could inhibit *E. coli* motility, reduce *E. coli*-associated inflammatory pathways and also downregulate bacterial production of the mutagenic toxin colibactin [118,119]. For this reason, gut microbiota modulation through SCFAs could be a promising strategy in preventing intestinal barrier disruption in CRC.

Based on these data, it is tempting to speculate that inhibition of specific bacterial species could be an adjuvant therapy in CRC. However, a clear cause–effect relationship between microbial colonization of CRC and tumor development has not yet been demonstrated [120], and the usefulness of antimicrobial therapy for the CRC treatment is made even more controversial by evidence showing an increased risk of CRC occurrence in patients exposed to oral antibiotics [121].

## 5. Gut–Vascular Barrier in Inflammatory Bowel Diseases

Inflammatory bowel diseases (IBDs) are characterized by the presence of an abnormal intestinal inflammation, arising from the complex interplay between genetic, individual and environmental factors [122,123,124].

Endothelial dysfunction participates in IBDs pathogenesis, since inflammation triggers the expression of endothelial adhesion molecules, increasing leukocyte recruitment in the affected mucosa [125,126]. In murine models of colitis, the increased expression of IFN-γ was able to drive mucosal damage and microvascular leakage [127]. In human umbilical vein ECs (HUVECs), IFN-γ was found to downregulate occludin expression, increasing vascular permeability, whereas pre-treatment with interleukin-10 (IL-10) was able to attenuate these effects [128]. It was recently shown both in HUVECs and DSS-induced colitis that IFN-γ could downregulate vascular endothelial (VE)-cadherin expression, whereas the administration of Imatinib, a tyrosine kinase inhibitor, was able to counteract this event, decreasing vascular permeability [127]. In experimental models of IBD, probiotics administration through *Bacillus* spores could reduce VCAM-1 and ICAM-1 serum levels, thus lowering mucosal leukocytes recruitment and also favoring endothelial homeostasis [129].

Moreover, the vascular endothelial growth factor (VEGF) pathway is upregulated in IBDs, as angiogenesis is a hallmark of active disease [130,131]. VEGF can stimulate in vitro expression of the PLVAP gene [132], and PV-1 is increased in patients with ulcerative colitis [133]. Notably, in BBB, VEGF is able to downregulate claudin-5 expression and induce vascular leakage [134]. These findings support the role of gut inflammation in causing GVB impairment in IBDs, but further studies are needed to understand its significance in IBDs pathogenesis and natural history.

## 6. Gut–Vascular Barrier in Celiac Disease

In celiac disease (CD), hypertransaminasemia is a common feature that correlates with the degree of duodenal damage [135]. The etiology of transaminase elevation is often multifactorial, and an increased intestinal permeability is a concurrent factor [136,137]. Spadoni et al. demonstrated that, despite adherence to a gluten-free diet, patients with CD and hypertransaminasemia had an increased intestinal expression of PV-1 [36]. De Leo et al. also conducted a retrospective study on a small population of CD patients with or without transaminase elevation; they showed that serum but not mucosal PV-1 was elevated in patients with CD and hypertransaminasemia not following a gluten-free diet, with a reversal after starting a gluten-free regimen. Interestingly, among controls, only IBD patients had an increased expression of PV-1 from mucosal origin, even if they did not show hypertransaminasemia. Thus, PV-1 serum measurement could represent a potential diagnostic tool for liver injury in CD. At the same time, the absence of mucosal PV-1 expression in CD patients led researchers to hypothesize a hepatic vascular origin of PV-1 rather than intestinal, in contrast with the work by Spadoni et al. [36,138]. Since PV-1 expression in liver sinusoidal endothelial cells is documented in vivo [139], further studies are needed to confirm the hepatic origin of serum PV-1 in celiac disease.

## 7. Gut–Vascular Barrier in Spondyloarthritis

Spondyloarthritis (SpA) association with IBDs has led to the concept of a gut–joint axis, in which environmental factors, genetic predisposition and gut dysbiosis contribute to intestinal and joint inflammation [140,141].

Adherent and invasive bacteria seem to be prevalent in gut specimens from patients with active ankylosing spondylitis (AS), and this correlates with the degree of TJ derangement. Among cultivable bacteria from ileal samples, Gram-negative *E. coli* and *Prevotella* were isolated. Interestingly, increased expression of PV-1, reduced expression of vascular junctional adhesion molecule-A (JAM-A), VE-cadherin and a discontinuous staining of endothelial occludin were observed in the ileum of AS patients, suggesting GVB impairment. In vitro exposure of endothelial cells to zonulin provoked a downregulation of VE-cadherin and occludin expression, suggesting a mechanism of GVB impairment mediated by the zonulin pathway in patients affected by AS [142].

## 8. Gut–Vascular Barrier and The Gut–Brain Axis

The gut and brain are connected by nervous system structures, and by biochemical and endocrine factors, with a mutual influence that configures the gut–brain axis [143,144,145]. The gut–brain axis dysfunction has been proposed as a model for mood disorders, and psychiatric and neurodegenerative disease development [146,147,148,149,150]. In IBDs, the association between intestinal symptoms and mood disorders has been attributed to an intestinally driven neuroinflammation [151,152].

Neuroinflammation has been documented as a concurrent cause of hepatic encephalopathy (HE) and neurological impairment in liver diseases [153,154]. In mice models of liver damage, systemic TNF-α signaling after bile duct resection has been associated with cerebral monocytes recruitment [155], whereas IL-6 signaling consequent to bile duct ligation has been associated with an increased hippocampal endothelial activation and sickness behavior [156]. Additionally, TGF-β signaling has been related to neuroinflammation and BBB dysfunction in murine models of HE [157,158], confirming the central role of a dysfunctional gut–brain axis in triggering neurological impairment during liver diseases [153,154].

The gut microbiota plays a pivotal role in promoting neuroinflammation [159]. An experimental study analyzed HE mechanisms in conventional and GF-cirrhotic mice, documenting a significant increase in pro-inflammatory cytokines in the cortex and cerebellum of conventional cirrhotic mice but not GF mice; moreover, a positive correlation between neuroinflammation, systemic inflammation and dysbiosis was observed in conventional cirrhotic mice [160]. In another study, GF mice colonization with stools of cirrhotic patients affected by HE, was associated with an increased expression of IL-1β and markers of microglial activation in the brain frontal cortex, while this was not observed in mice receiving FMT from healthy donors [161], yet confirming the role of dysbiosis in gut–brain axis dysfunction.

The gut microbiota influences the gut–brain axis at multiple levels, participating in the synthesis of neurotransmitters such as serotonin, dopamine, gamma-aminobutyric acid (GABA) and trace amines [162].

Bacteria also regulate stress responses and depression/anxiety disorders through the modulation of the hypothalamic–pituitary–adrenal axis [163], influence social behavior in neuropsychiatric diseases [164,165], promote microglia maturation and efficiency [166] and influence gut immune system activation in degenerative and neuro-immune diseases [167].

As previously mentioned, the GVB has several analogies with the BBB [39], which is extremely selective in molecules transport thanks to the presence of solid endothelial TJs [168]. On the other side, the blood–cerebrospinal fluid barrier (BCSFB), which regulates cerebrospinal fluid (CSF) synthesis in choroid plexus (CP), is more permeable to proteins, thanks to the presence of a fenestrated endothelium below a cuboidal epithelium [169].

The gut microbiota is a key regulator of BBB permeability, as mice from GF dams have a more permeable BBB in contrast to mice born from pathogen-free dams; furthermore, the lack of gut microbiota colonization causes brain damage secondary to vascular leakage in adult mice [170]. This phenomenon has been attributed to reduced expression of claudin-5 and occludin in the brain of adult GF mice, whereas butyrate administration restored occludin expression in the frontal cortex and hippocampus [170]. Butyrate, as well as propionate and acetate, is a SCFA derived from gut microbiota metabolism of dietary fibers; SCFAs are involved in maintaining intestinal barrier integrity, promoting TJs stability and exerting anti-inflammatory properties [171,172]. Butyrate and propionate are able to reduce VCAM-1 expression in HUVEC exposed to TNF-α [173]; in the same condition, butyrate, propionate and acetate are able to decrease IL-6 and IL-8 production, even if in a heterogeneous time-dependent manner [173]. Finally, it was reported that SCFAs can cross the BBB, inhibit the release of cytokines from microglial cells and reinforce brain endothelium in experimental studies [170,174]. For this reason, it is tempting to speculate that SCFAs are important contributors not only of the IEB stability, but also of GVB and endothelial homeostasis [24].

Recently, the presence of a brain choroid plexus vascular barrier (PVB), which regulates brain permeability in response to GVB damage, has been linked with the occurrence of mental disorders in patients with IBDs [133].

Differently from BBB, the endothelium of BCSFB is permeable to molecules till 70 kDa size [133]. Fenestrated ECs from BCSFB express the PLVAP gene, encoding for PV-1 [29,175]. PV-1 regulates leukocytes endothelial transmigration in vivo and in vitro, participating in the inflammatory process [176].

The CP is a site of immune cell trafficking, and modulates the passage of leukocytes and cytokines into CSF during neuroinflammatory conditions and in response to peripheral triggers [177,178]. Macrophages and T lymphocytes colonize the CP, and participate in immune response similarly to resident immune cells in gut lamina propria; structural analogies between the CP and gut–vascular barrier also include the presence of pericytes surrounding ECs, and a single epithelial layer covering the endothelial layer [179].

In their experimental work, Carloni et al. identified an increased expression of PV-1 in tissue specimens deriving from UC patients; secondly, they recreated a mouse model of dextran sodium sulfate (DSS)-induced colitis, observing an incremental PV-1 expression in intestinal ECs after DSS administration. DSS-induced colitis led to immune cell recruitment and microglia activation in the brain [133]. Intriguingly, the study demonstrated that the PVB promptly reacts to increased intestinal inflammation, self-limiting brain damage through upregulation of ECs sealing in a time-dependent manner. After the administration of high-molecular-weight (70-kDa) Cy7-conjugated dextran in mice, a precocious dextran extravasation in stromal CP and CSF at day 1 (T1) after treatment was observed, followed by a secondary and rapid shut-off of dextran leakage at day 3 (T3). This temporal modification of CP permeability was associated with an increased expression of PV-1 in CP ECs at T1 and a decrease at T3. At gene analysis, tissues from CP showed an upregulation of the PLVAP gene at T1 and a downregulation at T3, whereas an up-regulation of the Wnt/β-catenin pathway was observed at T1. Overall, these results provide evidence that the PVB regulates brain homeostasis reducing its permeability in response to intestinal inflammation via the Wnt/β-catenin pathway [133]. Given the profound implication of cognitive and social impairment during IBDs [180,181,182], the study also investigated whether modifications of PVB permeability could be responsible for behavioral changes during colitis [133]. Mice affected by colitis developed anxiety and impaired short-term episodic memory [183,184]. Wnt/β-catenin gain of function mice, characterized by a constitutive closure of PVB, still exhibited anxiety and memory impairment, suggesting that the interruption of gut–brain vascular axis could be responsible for IBDs-related mental deficits (Figure 2B) [133].

These studies enlighten the role of the PVB as a site of immune sensing and regulation between the brain and periphery, and a gatekeeper of brain health during inflammatory conditions; understanding the mechanisms of neuroprotection and immune regulation established by the PVB could help manage degenerative and infectious disorders. Nevertheless, considering the role of gut dysbiosis in neuroinflammation, microbiota-targeted therapies such as FMT or probiotics administration, which can increase SCFAs production and strengthen the gut barrier, might be used as adjuvants to reduce the risk of neurological impairment and counteract GVB–PVB dysfunction. However, PVBs shutting down during intestinal inflammation despite being a protective mechanism seems responsible for the impairment of social behavior in IBD, making GVB–PVB a double-edged sword and an insidious therapeutic target.

The role of the PVB as a site of immune cell trafficking is emphasized by the recognition of adhesion molecules in choroid plexus endothelium, other than epithelium [185].

ECs in CP intercept signaling from systemic circulation, including the GVB [179], participating in brain protection during inflammatory insults, as previously described. It would be of great interest to clarify whether inhibition of immune cells translocation at the endothelial side, without interruption of the GVB–PVB axis, could counteract intestinally associated neurological disorders and, more extensively, neuroinflammation in different disease settings.

## 9. Conclusions

In summary, considering the emerging role of the GVB in influencing the gut–brain axis and the gut–liver axis at different points, as well as its intriguing implications in the development of CRC, targeting the GVB seems to be a critical research field. The endothelium is the gatekeeper of vessels’ and tissues’ health, and plays a crucial role in different settings, from cardiovascular diseases to cancer. As emerged from several studies, ECs can acquire mesenchymal features losing cell-to-cell adhesion proteins and transforming into fibroblasts; this pathway contributes to tissue fibrosis but also cancer spreading, and is influenced by the gut microbiota. Therefore, understanding the GVB function is a research topic to be explored in the future for preventive, diagnostic, and prognostic purposes, especially for the personalization of therapeutic approaches.

## Figures and Tables

**Figure 1 ijms-24-01470-f001:**
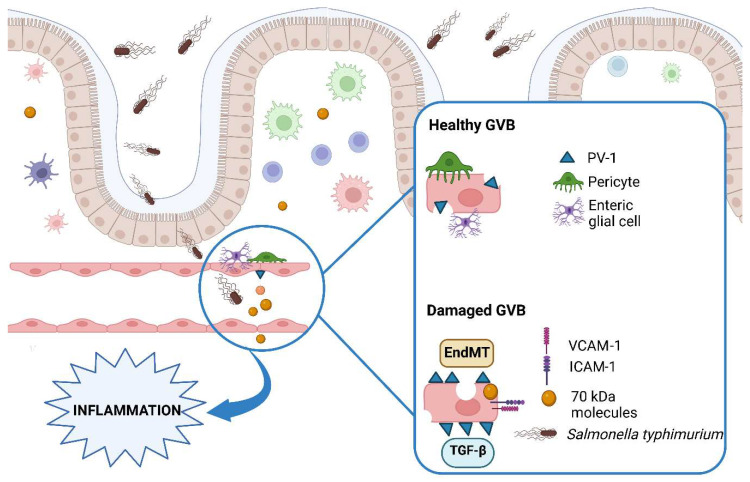
Gut–vascular barrier structure in health and inflammatory conditions. The GVB unit is composed of ECs surrounded by pericytes and enteric glial cells. ECs host small fenestrations, the permeability of which is regulated by the PV-1 protein and express adhesion molecules (VCAM-1, ICAM-1), participating in the inflammatory process. In the case of mucosal damage, as occurs during Salmonella infection, gut endothelial permeability is increased and the passage of larger molecules is permitted. EndMT and TGF-β signaling seem to be involved in this process. Thus, increased EC permeability represents a trigger for gut and systemic inflammation. Abbreviations: GVB: gut–vascular barrier; ECs: endothelial cells; PV-1: Plasmalemma vesicle-associated protein-1; VCAM-1: Vascular cell adhesion molecule 1; ICAM-1: Intercellular adhesion molecule 1; EndMT: endothelial to mesenchymal transition; TGF-β: Transforming growth factor beta. Created with BioRender.com.

**Figure 2 ijms-24-01470-f002:**
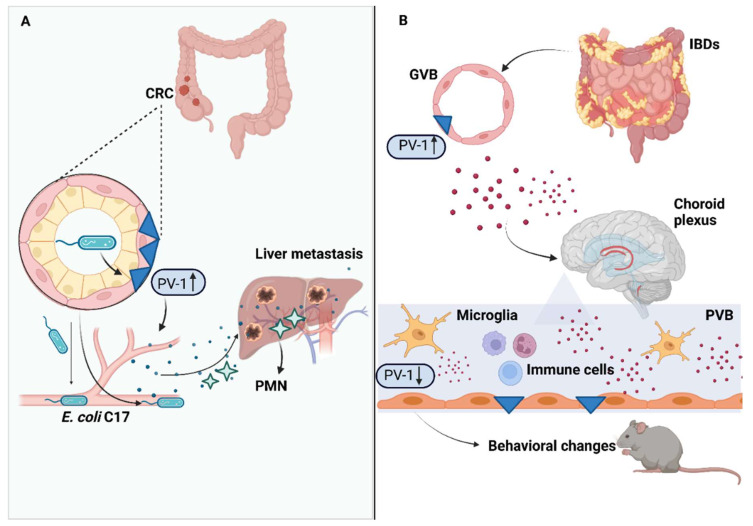
Influence of GVB disruption in colorectal cancer progression and in gut–brain axis function. Panel (**A**): In colorectal cancer, increased PV-1 expression is associated with disease progression. Microbiota may be a driver of this process, as bacterial dissemination of *E. coli* C17 from CRC lesions to the liver promotes the development of a premetastatic niche, favoring metastatic diffusion by disrupting the GVB. Panel (**B**): GVB damage occurs as a consequence of intestinal inflammation in IBDs and is associated with increased PV-1 expression. Intestinal inflammation drives immune cell activation in the brain. The PVB reacts to peripheral inflammation, increasing endothelial sealing to protect gut–brain axis homeostasis, a mechanism which has been linked to behavioral and cognitive impairment. Abbreviations: GVB: gut–vascular barrier; CRC: colorectal cancer; PMN: premetastatic niche; PV-1: Plasmalemma-vesicle associated protein-1; PVB: choroid plexus vascular barrier; IBDs: inflammatory bowel diseases. Created with BioRender.com.

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
