# Peer review of "The Gut–Vascular Barrier as a New Protagonist in Intestinal and Extraintestinal Diseases"

_ijms, 2023, doi:10.3390/ijms24021470_

Round 1

Reviewer 1 Report

This review paper is well written. However, there is some space for improvement. In most sections, the paper has descriptive cataloguing of past studies or information rather than a critical review of past studies. I would like to suggest critically scrutinizing the information from past studies and giving the author’s opinion. One suggested way authors can consider is to discuss each section at the end of that specific section and remove the discussion at the end of the paper. However, the final paragraph can make concluding remarks. 

Some minor edits suggested:

Line 62 – change object to objective

Line 71 to 75 – the sentence is hard to understand. Please consider rewriting. 

Line 140 – please check space.

Line 349 – check sentence structure – unnecessary has!

Line 469 – check sentence structure – unnecessary in!

Author Response

Q) This review paper is well written. However, there is some space for improvement. In most sections, the paper has descriptive cataloguing of past studies or information rather than a critical review of past studies. I would like to suggest critically scrutinizing the information from past studies and giving the author’s opinion. One suggested way authors can consider is to discuss each section at the end of that specific section and remove the discussion at the end of the paper. However, the final paragraph can make concluding remarks.

  1. A) We thank the Reviewer for this suggestion. We added a final remark at the end of chapter 3.1 (Nonalcoholic fatty liver disease and nonalcoholic steatohepatitis), chapter 3.2 (Alcoholic liver disease), chapter 3.3 (Cirrhosis), chapter 4 (Gut vascular barrier in colorectal cancer progression), chapter 8 (Gut vascular barrier and the gut-brain axis).

We also added the following studies:

-In chapter 3.1, we included two articles explaining possible non-invasive interventions for NAFLD treatment (10.3389/fnut.2021.716783; DOI: 10.1155/2021/2264737).

-In chapter 4.0, we included two articles (https://doi.org/10.1080/1040841X.2018.1481013 ; https://doi.org/10.1038/s41416-021-01665-7 ) that illustrate the actual challenges in considering the gut microbiota as a driver of CRC tumorigenesis.

-In chapter 8, we also included a paragraph (line 451-454) where we discuss a significant article from Carloni (doi: https://doi.org/10.1007/s00281-022-00955-3 ), that we also discussed at the end of this specific section. We also added an article from Figueiredo (doi: 10.1186/s12974-021-02370-1) explaining the role of choroid plexus endothelium as a site of immune cell trafficking.

Finally, as suggested, we changed the title of the final paragraph from “Discussion” to “Conclusion”, in which we included our final considerations.

We also changed our first figure (figure 1) since there were two wrong acronyms that we corrected, as well as in the text below the figure.

Q) Line 62 – change object to objective
A) Thanks for this suggestion, we corrected the word.

Q) Line 71 to 75 – the sentence is hard to understand. Please consider rewriting.
A) We have re-written the paragraph as suggested to make it more understandable. We also re-writed the paragraph 67-70 and 84-86 to make them more clear.

  1. Q) Line 140 – please check space.
    A) Spaced changed as suggested.
  2. Q) Line 349 – check sentence structure – unnecessary has!
  3. A) We corrected the sentence as suggested (line 390 in our version).

Q Line 469 – check sentence structure – unnecessary in!

  1. A) We deleted the sentence as the final chapter was re-written, according to the Reviewer suggestion.

Reviewer 2 Report

 You have to re-write the paragrafs dedicated to:

1. Liver disease

please consider the following article dedicated to liver protection exerted by a probiotic with bacillus spp by influencing ZO-1 as a TJ protein

Neag MA, Catinean A, Muntean DM, Pop MR, Bocsan CI, Botan EC, Buzoianu AD. Probiotic Bacillus Spores Protect Against Acetaminophen Induced Acute Liver Injury in Rats. Nutrients. 2020 Feb 27;12(3):632. doi: 10.3390/nu12030632. PMID: 32120994; PMCID: PMC7146158

2,Alcoholic Liver Disease paragraf  based on the follow article 

Martino, C., Zaramela, L.S., Gao, B. et al. Acetate reprograms gut microbiota during alcohol consumption. Nat Commun 13, 4630 (2022). https://doi.org/10.1038/s41467-022-31973-2

and search for the associations between SCFA production, E.Coli overgrowth and GVB dysfunction.

In the paragraf dedicated to GVB in IBD there are important conclusion that you could add regarding ICAM-1 and VCAM-1. I'm sorry that is my reserch and I'm not pushing of citation.

Catinean A, Neag MA, Krishnan K, Muntean DM, Bocsan CI, Pop RM, Mitre AO, Melincovici CS, Buzoianu AD. Probiotic Bacillus Spores Together with Amino Acids and Immunoglobulins Exert Protective Effects on a Rat Model of Ulcerative Colitis. Nutrients. 2020 Nov 24;12(12):3607. doi: 10.3390/nu12123607. PMID: 33255321; PMCID: PMC7760876.

Author Response

Q) 1. Liver disease

please consider the following article dedicated to liver protection exerted by a probiotic with bacillus spp by influencing ZO-1 as a TJ protein

Neag MA, Catinean A, Muntean DM, Pop MR, Bocsan CI, Botan EC, Buzoianu AD. Probiotic Bacillus Spores Protect Against Acetaminophen Induced Acute Liver Injury in Rats. Nutrients. 2020 Feb 27;12(3):632. doi: 10.3390/nu12030632. PMID: 32120994; PMCID: PMC7146158

A) As suggested by the Reviewer, we introduced the article at line 172-173 of our version.

Q) 2. Alcoholic Liver Disease paragraf  based on the follow article 

Martino, C., Zaramela, L.S., Gao, B. et al. Acetate reprograms gut microbiota during alcohol consumption. Nat Commun 13, 4630 (2022). https://doi.org/10.1038/s41467-022-31973-2

and search for the associations between SCFA production, E.Coli overgrowth and GVB dysfunction.

A) We thank the Reviewer for the suggested article of Martino and colleagues, which was added in chapter 3.2. We also added two more articles explaining the metabolism of alcohol and its implication in alcoholic liver disease (https://doi.org/10.1016/j.mehy.2020.109638 ; http://dx.doi.org/10.1053/j.gastro.2014.09.014 ).   We also modified a sentence at line 217.

Furthermore, in chapter 4 we illustrated the research from Bertocchi et al. which describes how E. Coli influences colorectal cancer progression through GVB damage. We added at the end of the chapter two articles (10.3390/antibiotics9080462 ; 10.4161/19490976.2014.969989 ) that further emphasize the implications of E. Coli in colorectal cancer with a particular mention of SCFAs role in contrasting this process.

Q) In the paragraf dedicated to GVB in IBD there are important conclusion that you could add regarding ICAM-1 and VCAM-1. I'm sorry that is my reserch and I'm not pushing of citation.

Catinean A, Neag MA, Krishnan K, Muntean DM, Bocsan CI, Pop RM, Mitre AO, Melincovici CS, Buzoianu AD. Probiotic Bacillus Spores Together with Amino Acids and Immunoglobulins Exert Protective Effects on a Rat Model of Ulcerative Colitis. Nutrients. 2020 Nov 24;12(12):3607. doi: 10.3390/nu12123607. PMID: 33255321; PMCID: PMC7760876.

A) Thank to the Reviewer for this suggestion, we included the reference in chapter 5.

Round 2

Reviewer 2 Report

excellent work. congratulations to the authors who made the suggested changes.